# High Entropy Borides Synthesized by the Thermal Reduction of Metal Oxides in a Microwave Plasma

**DOI:** 10.3390/ma16124475

**Published:** 2023-06-20

**Authors:** Bria Storr, Carolina Amezaga, Luke Moore, Seth Iwan, Yogesh K. Vohra, Cheng-Chien Chen, Shane A. Catledge

**Affiliations:** 1Department of Physics, University of Alabama at Birmingham, Birmingham, AL 35294, USA; bcstorr@uab.edu (B.S.); lmoore75@uab.edu (L.M.); iwanseth@uab.edu (S.I.); ykvohra@uab.edu (Y.K.V.); chencc@uab.edu (C.-C.C.); 2Department of Materials Engineering, Auburn University, Auburn, AL 36849, USA; cza0034@auburn.edu

**Keywords:** high entropy boride, ceramics, boro/carbothermal reduction

## Abstract

Metal oxide thermal reduction, enabled by microwave-induced plasma, was used to synthesize high entropy borides (HEBs). This approach capitalized on the ability of a microwave (MW) plasma source to efficiently transfer thermal energy to drive chemical reactions in an argon-rich plasma. A predominantly single-phase hexagonal AlB2-type structural characteristic of HEBs was obtained by boro/carbothermal reduction as well as by borothermal reduction. We compare the microstructural, mechanical, and oxidation resistance properties using the two different thermal reduction approaches (i.e., with and without carbon as a reducing agent). The plasma-annealed HEB (Hf_0.2_, Zr_0.2_, Ti_0.2_, Ta_0.2_, Mo_0.2_)B_2_ made via boro/carbothermal reduction resulted in a higher measured hardness (38 ± 4 GPa) compared to the same HEB made via borothermal reduction (28 ± 3 GPa). These hardness values were consistent with the theoretical value of ~33 GPa obtained by first-principles simulations using special quasi-random structures. Sample cross-sections were evaluated to examine the effects of the plasma on structural, compositional, and mechanical homogeneity throughout the HEB thickness. MW-plasma-produced HEBs synthesized with carbon exhibit a reduced porosity, higher density, and higher average hardness when compared to HEBs made without carbon.

## 1. Introduction

High entropy transition metal borides are solid solutions that typically contain five or more metals mixed in equimolar (or near equimolar) concentrations. They have gained interest largely due to their thermal/mechanical properties and potential applications [1,2]. They are known to be a part of the ultra-high-temperature ceramics (UHTCs) group [3]. Compared to individual transition metal diborides, high entropy borides (HEBs) exhibit useful properties such as high melting points, good oxidation resistance, and high hardness [4,5]. Such properties allow these materials to be excellent candidates for performance under extreme conditions. Applications include armors, high-temperature wear-resistant parts, cutting tools, and hypersonic vehicles [6,7]. HEBs are typically synthesized via convective heating in vacuum ovens or furnaces, followed by a final densification step to achieve near-theoretical density. This final step is often performed by the Field-Assisted Sintering Technique (FAST), also known as Spark Plasma Sintering (SPS) [8].

Boro and carbothermal reductions are metallurgical processes for reducing metal oxides to form transition metal borides and other ceramics. This reduction typically occurs between 1200 and 1550 °C. In this work, boron carbide and carbon black are used for boro/carbothermal reductions (BCTR). The reduction of transition metal oxides to form HEBs leads to volatile carbon monoxide that can be pumped out via the vacuum system. Volatile boron oxides can also form in this process, which typically necessitates the use of excess boron carbide precursor, the amount of which is experimentally optimized [9]. In contrast, the borothermal reduction (BTR) approach has the benefit of not needing carbon in the precursor mixture, while still forming volatile boron oxides. Important parameters leading to successful metal oxide reduction include reaction temperature, particle size, ratio between metal oxides and reducing agents, inert gas flow rate, and pellet size [10]. Carbon black is a good microwave absorber due to its high dielectric loss tangent. By interacting with the electric field in the microwave process, the ramp up time to the reaction temperature is rapid [11]. In the microwave-induced plasma process, metal oxide reduction and subsequent formation of the HEB solid solution is performed as a single annealing step within 1 h at 2000 °C.

Individually, both microwave (MW) sintering and plasma sintering are known to create conditions for rapid processing, enhanced mechanical properties, and refinements in microstructure. Convectional heating cannot replicate the targeted heating method of plasma sintering. The MW plasma effectively transfers heat to the sample to allow high heating rates [12]. The MW-induced plasma technique reduces synthesis time by enhancing reaction kinetics, provides efficient energy transfer, and is considered eco-friendly. The rapid MW heating in this process lends itself to less processing time at elevated temperatures and can therefore help mitigate particle coarsening and allow more effective densification in subsequent sintering steps [7,11,13,14]. In the present study, we compare boro/carbothermal and borothermal reductions from a microwave plasma process. Structural, mechanical, and thermal properties are characterized for different HEB cross-sections in order to evaluate sample homogeneity. In the present study, we compare boro/carbothermal and borothermal reductions from a microwave plasma process. Structural, mechanical, and thermal properties are characterized for different HEB cross-sections in order to evaluate sample homogeneity. By investigating the two approaches for metal oxide reduction using MW plasma, we aim to highlight the differences in properties as well as the benefits and disadvantages of each process.

## 2. Materials and Methods

The synthesis of near-equimolar high entropy transition metal borides was achieved using MW–induced plasma and two different reduction methods: boro/carbothermal reduction (BCTR) and borothermal reduction (BTR). The precursor powders included the metal oxides: hafnium oxide (HfO_2_, 99%, 325 mesh; Alfa Aesar, Ward Hill, MA, USA), zirconium oxide (ZrO_2_, 99+%, 325 Mesh; Alfa Aesar), titanium oxide (TiO_2_, 99.6%, 325 Mesh; Alfa Aesar), tantalum pentoxide (Ta_2_O_5_, >60 mesh, 99%; Alfa Aesar), and molybdenum oxide (MoO_3_, <325 mesh, 99.95%; Alfa Aesar). Reducing agents for BCTR included carbon black (C, 99.99+%; Alfa Aesar) and boron carbide (B_4_C, purity 99+%, 325 mesh; Alfa Aesar). In the case of BTR, no carbon was used, and the only boron source was pure boron powder (B, 98%, Thermo Fisher Scientific, Waltham, MA, USA). Figure 1 shows the stoichiometric equations for both metal oxide reduction methods explored in this work. Balanced chemical equations were used to measure the appropriate amount of precursors to produce the near equimolar HEB, given as (Hf_0.2_, Zr_0.2_, Ti_0.2,_ Ta_0.2,_ Mo_0.2_)B_2_. This configuration is ideal due its high entropy forming ability of 207 (eV/atom)^−1^ and theoretical hardness of approximately 33 GPa [15].

The precursor powders of the five metal oxides and reducing agents—boron, carbon, or boron carbide—were first blended using a high-energy ball mill (Spex 8000M, Spex SamplePrep Inc., Metuchen, NJ, USA) for 6 h with a tungsten carbide (WC) vial. The precursor was dry milled for 2 h with WC balls and wet milled for 4 h with zirconia balls in acetone. Due to the expected loss of boron during the annealing process, excess B_4_C or boron was added for either BCTR or BTR, respectively. The blended powder was dried at room temperature in a vacuum oven, then passed through a 200-mesh sieve before being pressed into a 5mm diameter pellet at a uniaxial pressure of 50 MPa.

The pellet was placed on a molybdenum platform in the microwave plasma chamber (Wavemat Inc., Plymouth, MI, USA). The chamber was pumped to a base pressure of 1.1 Pa before a feed gas mixture (500 sccm Ar, 80 sccm H_2_) was added with subsequent plasma ignition using a 2.45 GHz magnetron. It was found that the small addition of H_2_ (in addition to argon) helped to stabilize and concentrate the sample near the sample. A microwave power of 1.3–1.5 kW and chamber pressure of 2.4 × 10^4^–3.3 × 10^4^ Pa was used to rapidly heat up the pellet (70–100 °C/min in this system) to 2000 °C, as also described in our previous works [15,16]. The precursor pellets were MW-plasma annealed using either BCTR or BTR at 2000 °C for 1 h.

As a means of comparison, a BCTR-HEB (Hf_0.2_, Zr_0.2_, Ti_0.2_, Ta_0.2_, Mo_0.2_)B_2_ was pressed into a 10 mm pellet, MW-plasma-annealed at 2000 °C, and then FAST-sintered in a Dr. Fritsch model DSP-507 press (Fellbach, Germany). The FAST sintering was performed at 2000 °C at 45 MPa for a duration of 10 min in a 12.7 mm graphite die.

The sample densities were measured via the Archimedes method. In order to calculate the theoretical densities, the lattice parameters were determined based on the XRD patterns. The quantitative analysis was performed by Rietveld refinement using the X’Pert HighScore Plus (version 4.8) software. The crystalline structure was characterized by X-ray diffraction (XRD, Panalytical Empyrean, Malvern, Worcestershire, UK). XRD scans were collected with Cu Kα radiation (λ = 1.54186 Å) at 45 kV and 40 mA, with 12.75 s per step and a 0.0131° step size.

Oxidation resistance was tested at a high temperature using a thermogravimetric analyzer (TGA, Linseis, Selb, Germany). The MW plasma samples remained in pellet form during plasma annealing and when evaluating oxidation resistance via TGA. The BCTR + FAST sample was cut in a cube (2 mm × 2.05 mm × 3.1 mm) for oxidation resistance testing. The weight range of all the samples was within 70–80 mg. Individually, the samples were exposed to controlled airflow with a temperature ramp-up rate of 2 °C/min to 1300 °C.

SEM/EDX images were collected using a Quanta FEG 650 at an accelerating voltage of 15 kV. The HEBs’ surfaces were grinded using Dia-Grid discs (9 micron to 125 micron; from Allied High Tech Products, Inc., Cerritos, CA, USA), polished using diamond slurry (9 micron to 1 micron), and then ultrasonically cleaned in isopropyl alcohol. Vickers hardness measurements were performed using a Phase II+ 900-390A Micro Vickers Hardness tester (TEquipment, Long Branch, NJ, USA). An average of 10 indents were made per sample using a load of 200 gf with a dwell time of 15 s.

Density functional theory (DFT) calculations were performed with the Vienna Ab initio Simulation Package (VASP) [17,18] using the Perdew–Burke–Ernzerhof exchange correlation functional (PBE-GGA) [19] and the projector augmented wave (PAW) method [20,21]. A 450 eV energy cutoff for the wavefunction expansion and a k-grid mesh of at least 8000 k-points per reciprocal atom (KPPRA) were utilized. The convergence criteria for self-consistent electronic and structural relaxation calculations were set to 10^−6^ eV and 10^−2^ eV per angstrom, respectively. The high-entropy alloying effect was simulated by the technique of special quasi-random structure (SQS) [22], which, in turn, was generated by Monte Carlo samplings using the Advanced Theoretical Alloy Toolkit (ATAT-mcsqs) [23]. The elastic constants (*C_ij_*) computed by VASP for each SQS structure were utilized to evaluate the mechanical properties, such as the bulk (B) and shear (G) moduli, using the Voight–Reuss–Hill approximation [24,25,26]. The resulting B and G values then enabled the estimation of Vickers hardness using Tian’s model [27].

## 3. Results and Discussion

### 3.1. X-ray Diffraction

Upon forming the HEB (Hf_0.2_, Zr_0.2_, Ti_0.2_, Ta_0.2_, Mo_0.2_)B_2_, the pellets were crushed into a powder for XRD analysis. As shown in Figure 2, the major phase for all samples is identified as a hexagonal AlB_2_-type structure, characteristic of HEBs [28]. Compared to BTR, the BCTR sample (made using carbon-reducing agents) resulted in a slight shift of XRD peaks to lower 2-theta (higher d-spacing). Table 1 shows the corresponding increase in lattice constants and unit cell volume for the BCTR sample. Due to the larger unit cell volume for BCTR and BCTR + FAST samples, the theoretical density (determined from the measured lattice constants) is lower than that of the BTR sample. The BCTR sample had a higher relative density (determined using Archimedes method) compared to that of BTR. Regardless of which reduction process was used, the dominant phase is associated with the hexagonal HEB structure.

Crystallite sizes and microstrains were estimated from XRD using the Williamson–Hall method via the uniform deformation approximation [29,30] (Equations (1) and (2)):(1)$$D=\frac{k\lambda }{\beta cos\;\theta }$$
(2)$$\beta cos\;\theta =\frac{k\lambda }{D}+4ɛsin\;\theta$$
where *D* corresponds to the crystallite size, ε is the microstrain, *β* is the full-width half- maximum value, *k* is the shape factor, λ is the X-ray wavelength (1.5406 Å), and *θ* is the diffraction angle at which the XRD peaks occurs. Ten data points (XRD peaks) were used in the line of best fit.

Given the measurement uncertainties in Table 2, the values for crystallite size and microstrain are statistically insignificant between the three synthesis methods. However, the BTR method resulted in the smallest average crystallite size and the highest average microstrain. Increased peak widths, along with decreased d-spacings, are evident in Figure 2 for the BTR sample when compared to the BCTR and (BCTR + FAST) samples. The addition of FAST to the BCTR method resulted in a 56% increase on the average crystallite size compared to BCTR alone. This is expected since the continued high temperature sintering employed by FAST is expected to lead to grain growth [31,32,33,34,35].

### 3.2. Oxidation Resistance

Figure 3 shows oxidation resistance measurements (sample weight gain vs. temperature). A commercial diboride, TiB_2_, was used as a control and was found to double its initial weight from room temperature to 1300 °C. An almost imperceptible difference was observed when comparing the two HEB samples made via the BCTR and BTR reduction methods. The difference in oxidation behavior corresponds to only about 0.6% between the two methods. Additionally, both BCTR and BTR reduction methods resulted in only modest weight gains of a few percent in the range of 700 °C to 1300 °C, with corresponding final weight gains of 13.1% and 13.6%, respectively. However, the FAST-synthesized HEB sample revealed excellent oxidation resistance with a weight gain of only 3.0% over the entire range of temperature. This result is comparable to other HEB oxidation resistance studies for samples sintered using FAST [36,37]. However, one benefit of using the metal oxide reduction process is that less contamination can occur from the WC vial during ball milling as a result of softer precursor materials used. In addition, the metal-oxide precursors are less expensive than the diboride precursors.

### 3.3. HEB Microstructure

As shown in the schematic of Figure 4a, the plasma contacts the upper portion of the pellet more than the lower portion. HEB composition, porosity, and mechanical properties were investigated for sample cross-sections prepared near the top and near the bottom of the pellets.

Figure 4b,c are optical microscopy cross-sections of the HEB pellets that correspond to the XRD patterns presented in Figure 2. In contrast to the BCTR and BTR pellets, the BCTR + FAST pellet shows almost no surface porosity, consistent with the 99.4% relative density measured for this pellet. After the MW plasma process, the pellets take on a truncated cone shape, suggesting more plasma-initiated reduction occurring near the top vs the bottom of the pellet. This is revealed in Figure 4d as before/after photographs. The difference in porosity for all the pellets from top to bottom can be explained by a temperature gradient with enhanced heating and densification in the upper portion of the pellet from the MW plasma. This leads to higher porosity near the pellet bottom.

The BCTR method results in reduced porosity compared to that of the BTR method. This can be explained by the relative amounts of boron oxide outgassed during the MW plasma process. As temperature increases, BCTR pellets generate boron oxide and carbon monoxide, while BTR pellets generate boron oxide without carbon byproducts. It is expected that BTR generates more boron oxide than BCTR since some of the oxygen in BCTR escapes from carbon monoxide. Volatilization of boron oxide from the surface occurs rapidly above 1400 °C. Studies on diborides have shown that the volatilization of B_2_O_3_ near the surface leads to the formation of pores, leaving metal oxides (particularly ZrO_2_ and HfO_2_) behind [38,39]. In our processes, the residual metal oxides are effectively converted by appropriate amounts of added excess reducing agents.

The difference in porosity between the two reduction methods can also be seen in SEM images after polishing the top surface of the HEB pellets (Figure 5). Regions of dark contrast in the secondary electron images (upper left panel for (a) and (b)) are identified as pores. The EDX elemental maps in Figure 5 reveal the individual elements Hf, Zr, Ti, Ta, Mo to be homogeneously distributed for both metal oxide reduction methods. The EDX maps are consistent with equimolar HEB structures, suggested by the XRD in Figure 2.

### 3.4. Hardness

The measured Vickers hardness of MW-plasma-synthesized HEBs are shown in Figure 6. The BCTR-HEB consistently presents a higher hardness than BTR-HEB, for both the top and bottom cross-sections investigated. The Vickers hardness for the BCTR-HEB sample was measured to be 38 ± 4 GPa (top cross-section) and 31 ± 7 GPa (bottom cross-section). In comparison, the hardness for the BTR-HEB sample was measured to be 28 ± 3 GPa (top cross-section) and 19 ± 7 GPa (bottom cross-section). There is a larger standard deviation in hardness for the bottom side of the pellet due to the more porous nature of this cross-section, resulting from the reduced exposure of this surface to the plasma. Micro-hardness measurements are known to be affected by sample porosity. As porosity increases (and relative density decreases), the material under the indenter can deform more readily, causing a reduction in apparent hardness [40]. This slight increase in density of about 1.2% with the BCTR approach, paired with fewer pores, results in higher hardness. The measured average hardness of 38 ± 2 GPA for the BCTR + FAST sample is comparable to that of the BCTR sample. The higher densification for the BCTR + FAST sample resulted in a lower standard deviation for the hardness measurement (*n* = 10 indents).

Finally, we benchmarked our experimental hardness measurements against theoretical values. The theoretical hardness for the HEBs under study has been computed to be approximately 33 GPa in our prior work [15], using a partial occupation method (POCC) of small 15-atom unit cells. Here, we performed additional DFT calculations using SQS structures of varying size and shape up to 150 atoms (see Figure 7a). Figure 7b shows that the corresponding hardness values computed with different SQS supercells have also essentially converged to a value of ~33 GPa. Therefore, the experimental hardness values of ~28–38 GPa, measured on the top cross-sections for the BCTR and BTR samples, were reasonable and also consistent with our simulations. Meanwhile, we note that the hardness of a material in general can be attributed to intrinsic and extrinsic effects. The intrinsic effect relies on the underlying chemical composition, atomic bonding, and crystal structure. The extrinsic effect depends on the surface morphology, microstructure, grain size, etc. Our DFT calculation mainly accounts for the microscopic intrinsic effect, and it cannot directly address the change in hardness between the BTR and BCTR samples, which is a mesoscopic extrinsic effect due to porous microstructure.

Overall, both reduction methods explored in this study have their advantages. BCTR results in a less porous material than BTR since it has fewer volatile boron oxide products, which would otherwise lead to pores due to the decreased outgassing of the volatile boron oxide. BCTR has a slight increase in density. This slight increase in density combined with fewer pores results in a higher apparent hardness. BTR’s main advantage is that there is no risk of carbon contamination, while maintaining good oxidation behavior comparable to the BCTR process. It may be best to follow this method with a field-assisted sintering technology (FAST), also known as spark plasma sintering. The combined BCTR + FAST approach may lead to optimal HEB processing since the BCTR step already creates the high entropy, predominantly single-phase material, while subsequent FAST processing may allow further densification at lower temperatures to minimize grain growth. This is something that will need to be confirmed in future work.

## 4. Conclusions

High entropy boride (Hf_0.2_, Zr_0.2_, Ti_0.2_, Ta_0.2_, Mo_0.2_)B_2_ powders were prepared by both boro/carbothermal and borothermal reduction via microwave-induced plasma annealed at 2000 °C. The XRD spectra for both reduction methods reveal a predominantly single-phase hexagonal AlB_2_ structure. Different degrees of surface porosity were observed between the two reduction methods. Cross-sections with more exposure to the plasma and carbon-based precursors (BCTR reduction method) resulted in a less porous surface. The highest hardness measured was 38 GPa for the top cross-section of BCTR-HEB, while the lowest measured hardness was 19 GPa for the bottom cross-section of BTR-HEB. The hardness measurements were consistent with first-principles simulations, showing a theoretical hardness value of ~33 GPa. The results of microstructure characterization and mechanical testing suggest that high entropy diboride ceramics can have a higher sintered density and a higher hardness when synthesized via the boro/carbothermal reduction method paired with MW plasma annealing.

## Figures and Tables

**Figure 1 materials-16-04475-f001:**
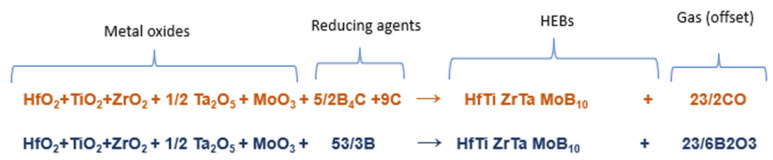
The stoichiometry equation for BCTR (orange; first row) and BTR (blue; second row).

**Figure 2 materials-16-04475-f002:**
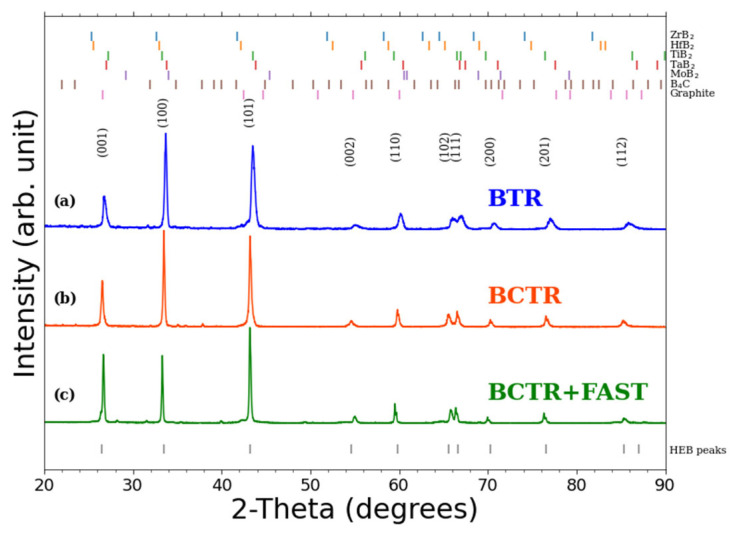
Powder XRD pattern of the HEB (Hf_0.2_, Zr_0.2_, Ti_0.2_, Ta_0.2_, Mo_0.2_)B_2_ after being annealed via MW plasma at 2000 °C: (**a**) BTR-HEB (blue) synthesized without carbon; (**b**) BCTR-HEB (orange) synthesized with carbon; (**c**) BCTR + FAST-HEB (green) synthesized from a mixture of five commercial diborides.

**Figure 3 materials-16-04475-f003:**
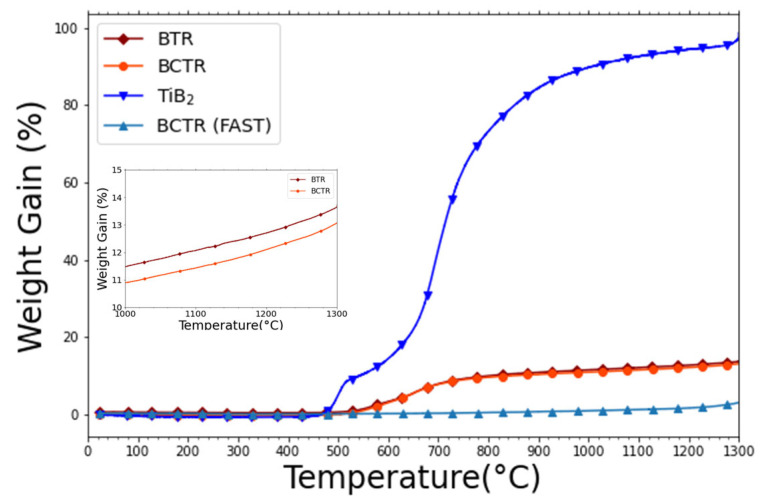
Weight gain vs. temperature as a measure of oxidation resistance. The samples include a commercial diboride TiB_2_ (blue; triangle) and the MW-plasma-synthesized HEBs: BCTR (orange; circle) and BTR (brown; diamond). The insert shows the plasma-annealed HEBs in the range 1000–1300 °C.

**Figure 4 materials-16-04475-f004:**
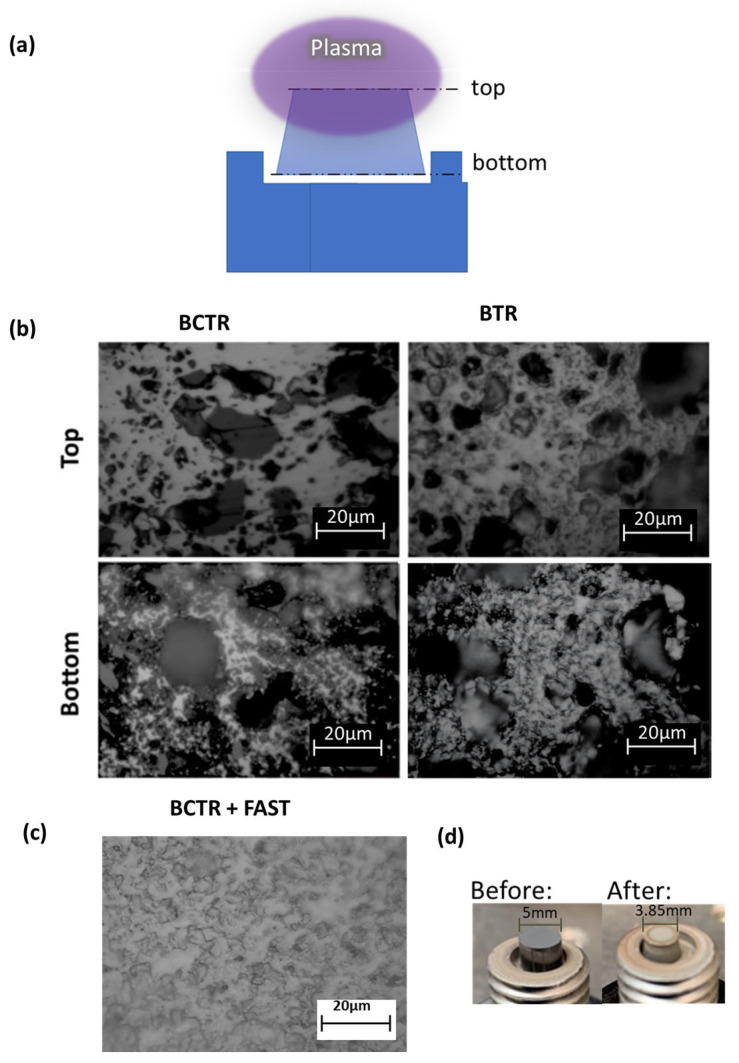
(**a**) A schematic of the pellet in the later stages of the MW plasma annealing process. Optical microscopy images (**b**) taken from the top and bottom sections of both the BTR and BCTR pellets, and (**c**) taken from the top surface of the BCTR + FAST pellet. (**d**) Image of the pellet before and after plasma annealing.

**Figure 5 materials-16-04475-f005:**
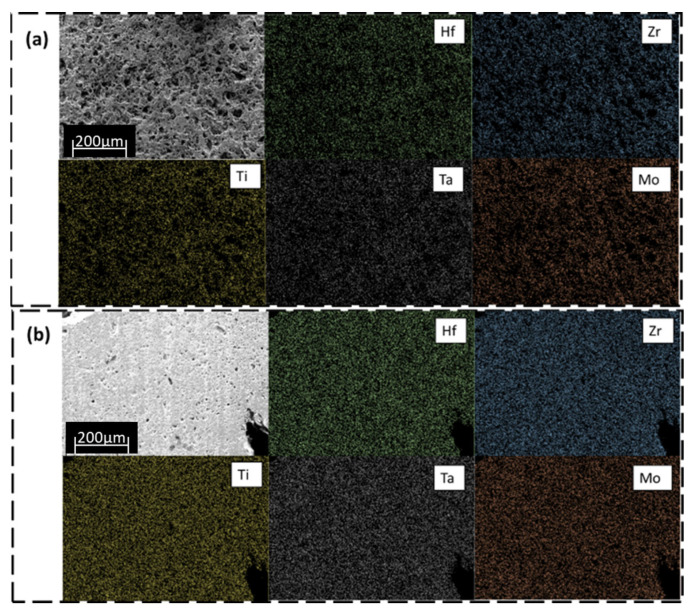
EDX mapping images of the polished surface of annealed (Hf_0.2_, Zr_0.2_, Ti_0.2_, Ta_0.2_, Mo_0.2_)B_2_ from (**a**) BTR and (**b**) BCTR. The upper left image with the scale is a secondary electron image.

**Figure 6 materials-16-04475-f006:**
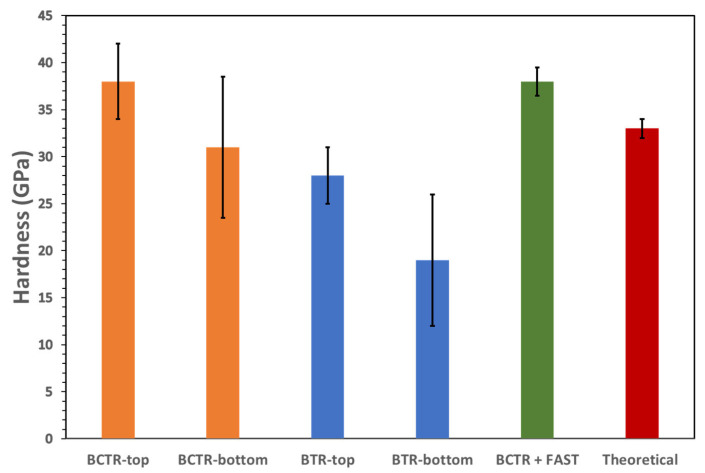
Vickers hardness of BCTR (orange), BTR (blue), and BCTR + FAST (green) at a load of 200 gf for 15 s. The hardness calculated from density functional theory (~33 GPa) is also shown (red).

**Figure 7 materials-16-04475-f007:**
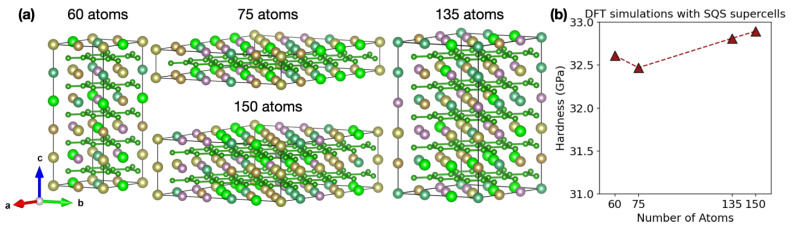
(**a**) Special quasi-random structures (SQSs) for density functional theory (DFT) calculations. The small green balls are boron atoms, and larger balls are transition metals. (**b**) The theoretical Vickers hardness values from the DFT and SQS simulations for different supercells were ~33 GPa, which was consistent with the experiments. The dashed red line serves as a visual guide.

**Table 1 materials-16-04475-t001:** Summary of measured material parameters for the HEB sample composition: (Hf_0.2_, Zr_0.2_, Ti_0.2_, Ta_0.2_, Mo_0.2_)B_2_.

Synthesis Method	Excess Reducing Agents	Measured Lattice Constants		Unit Cell Volume	Theoretical Density *
	(wt%)	a (Ǻ)	c (Ǻ)	(Ǻ)^3^	(g/cm^3^) (Relative Density)
BCTR	9% (B_4_C)	3.09023 (6)	3.35918 (6)	27.78	8.40 (93.7%)
BTR	10% (boron)	3.07890 (6)	3.34133 (6)	27.43	8.51 (92.5%)
BCTR + FAST	-	3.09901 (4)	3.33501 (4)	27.73	8.41(99.4%)

* The theoretical density is calculated from measured lattice parameters, determined using Rietveld refinement.

**Table 2 materials-16-04475-t002:** The crystallite size and microstrain for the BCTR, BTR, and BCTR + FAST as determined using the Williamson–Hall method.

Synthesis Method	Crystallite Size (nm)	Microstrain ɛ (×10^−3^)
BCTR	41.6 ± 17.9	1.26 ± 0.65
BTR	28.5 ± 10.5	1.69 ± 0.83
BCTR + FAST	64.9 ± 25.5	1.12 ± 0.39

## Data Availability

The data that support the findings of this study are available from the corresponding author upon reasonable request.

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
