# Peer review of "High Entropy Borides Synthesized by the Thermal Reduction of Metal Oxides in a Microwave Plasma"

_materials, 2023, doi:10.3390/ma16124475_

Round 1

Reviewer 1 Report

1. The authors should provide the macroscopic morphology of the samples in the paper.

2. The authors should further add the reasons for the difference in hardness between BCTR and BTR methods.

3. The scale in Figure 4(b) is too small and there are missing cases.

4. the authors should further explain the reasons for the reduced porosity of the samples prepared by the BCTR method

5. The authors performed the calculation and verification of hardness using the first nature principle, and in fact the authors could have performed calculations such as DOS to further explain the reasons for the change in hardness.

6. The font and scale in the picture should get unified, e.g. Delete the meaningless scale in the upper part of the picture.

Reviewer 2 Report

It is a correct scientific work about high entropy borides produced by thermal reduction of metal oxides in a microwave plasma. Nevertheless, I recommend a major revision.

XRD analysis: Which method apply the authors to determine the crystallographic parameters?

Likewise, it is necessary to add an error to the calculated lattice parameters.

I recommend improving the XRD discussion. It is possible to determine the crystalline size and the microstrain with Rietveld refinement or with linear methods such as Williamson-Hall.

Figure 7. The authors provide ideal atomic distributions. From the improved XRD analysis is possible to check (with the microstrain) the importance of the crystallographic parameters and to improve the discussion of this pare density functional theory

It is known that ball milling favors contamination from the milling tools. Please, check carefully to provide final composition.

Minor editing of English language required

Reviewer 3 Report

The presented work is of particular interest in the field of materials science. While reading the text of the manuscript, the reviewer had some remarks and comments.

1) The purpose of the study is missing;

2) Why was such a complex composition of the solid solution containing hafnium and zirconium chosen? Why is such a composition interesting from the point of view of high-entropy borides?

3) The title of section 3 suggests discussion. But the authors present only experimental data without discussing the results.

Why was TiB2 chosen as a control sample in measurements of oxidation resistance?

Based on what analysis was it concluded that the selected metals in the solid high-enthalpy boride are in the ratio 0.2:0.2:0.2:0.2:0.2? The authors showed mapping data, but not element ratios. What results were obtained regarding the ratio of metals?

How do the authors explain the difference in the porosity of the samples obtained by the two methods?

Is the porosity of the samples the same at the top and bottom?

4) The authors do not give explanations why one method is better than another? Please list the advantages and disadvantages of each method.

In addition, there are some other minor remarks.

Page 3 line 136: check the correct designation of the type structure;

Page 3 line 138: extra dot before number

Figure 6 is best represented as a histogram.

The submitted comments do not reduce the level of the results obtained and the answers to them will increase the readability of the material presented.

Round 2

Reviewer 2 Report

The authors modify the manuscript taking into account the comments of the referees.
Tha quality and soundness of the manuscript has been improved.

Minor editing of English language required

Author Response

Thank you. We have proof-read for minor editing of English.

Reviewer 3 Report

Thanks to the authors for the detailed responses to the comments. This increased the readability of the article.

Author Response

Thank you for acknowledging that we have satisfied the reviewer comments.